# Efficient Exact Verification of Binarized Neural Networks

**Kai Jia**
MIT CSAIL
32 Vassar St, Cambridge, MA 02139
jiakai@mit.edu

**Martin Rinard**
MIT CSAIL
32 Vassar St, Cambridge, MA 02139
rinard@csail.mit.edu

## Abstract

Concerned with the reliability of neural networks, researchers have developed verification techniques to prove their robustness. Most verifiers work with real-valued networks. Unfortunately, the exact (complete and sound) verifiers face scalability challenges and provide no correctness guarantees due to floating point errors. We argue that Binarized Neural Networks (BNNs) provide comparable robustness and allow exact and significantly more efficient verification. We present a new system, EEV, for efficient and exact verification of BNNs. EEV consists of two parts: (i) a novel SAT solver that speeds up BNN verification by natively handling the reified cardinality constraints arising in BNN encodings; and (ii) strategies to train solver-friendly robust BNNs by inducing balanced layer-wise sparsity and low cardinality bounds, and adaptively cancelling the gradients. We demonstrate the effectiveness of EEV by presenting the first exact verification results for $\ell_\infty$-bounded adversarial robustness of nontrivial convolutional BNNs on the MNIST and CIFAR10 datasets. Compared to exact verification of real-valued networks of the same architectures on the same tasks, EEV verifies BNNs hundreds to thousands of times faster, while delivering comparable verifiable accuracy in most cases.

## 1   Introduction

Deep Neural Networks (DNNs) have achieved impressive success in many applications including image understanding, speech recognition, natural language processing, and game playing [26]. Unfortunately, DNNs exhibit unexpected or potentially dangerous behavior due to limited robustness [56].

In response, researchers have developed techniques that attempt to verify that a DNN satisfies a robustness specification [34, 20, 57]. However, scalability is a formidable challenge for exact verifiers [10]. For example, MIPVerify [57] needs hundreds of seconds to verify the robustness of a network whose inference on CPU takes only a few milliseconds, even though the network is trained to be solver-friendly [61]. Additionally, floating point errors make any correctness guarantees effectively unobtainable for exact verification of nontrivial real-valued neural networks [32].

Binarized Neural Networks (BNNs) comprise an attractive alternative to real-valued neural networks. BNNs exhibit significant speed gain and energy savings during inference [29, 47, 42] while achieving competitive accuracy on challenging datasets [7]. Because of the absence of floating point arithmetic, BNNs can also support exact verification [13, 43]. To date, however, BNN verification has exhibited even worse scalability than real-valued neural network verification [44].

We present a new system, EEV, for exact and efficient verification of BNNs. EEV incorporates a novel SAT solver tailored for BNN verification and new training strategies that enhance the robustness and verification efficiency of the trained BNNs. We use EEV to verify the robustness of BNNs against input perturbations bounded by the $\ell_\infty$ norm. Our experimental results show that, compared to exact verification of robustly trained real-valued networks with the same architectures, EEV delivers several

orders of magnitude faster verification of BNNs with comparable verifiable accuracy in most cases. This paper makes the following contributions:

1. We incorporate native support for reified cardinality constraints into a SAT solver, which improves the performance of BNN verification by more than a factor of one hundred compared to an unmodified SAT solver (Section 4.2).

2. We identify that sparse weights induced by ternary quantization [44] cause unbalanced sparsity between layers of convolutional networks, which leads to high verification complexity despite sufficient overall sparsity. We propose a new strategy, BinMask, which produces more balanced sparsity. Our system verifies BNNs trained with BinMask two to five orders of magnitude faster than BNNs trained with ternary quantization (Section 5.1).

3. We further reduce the verification complexity of BNNs by introducing a regularizer that induces lower cardinality bounds, which leads to an additional speedup of up to thousands of times (Section 5.2).

4. We find that directly applying the PGD training algorithm [38] does not induce verifiably robust BNNs. We propose adaptive gradient cancelling to train robust BNNs (Section 6).

5. We present the first exact verification of robustness against $\ell_\infty$-norm bounded input perturbations of convolutional BNNs on MNIST and CIFAR10, and compare the results with real-valued networks. EEV verifies exact robustness of BNNs between 338.28 to 2440.65 times faster than a state-of-the-art exact verifier on real-valued networks with the same architecture, while delivering comparable verifiable accuracy in most cases (Table 4).

## 2   Background and related work

DNN exact (i.e., sound and complete) verification checks whether a DNN satisfies a specification, which, for example, can require the network to give robust predictions against input perturbations. Exact verifiers aim to provide either a proof of satisfaction or a counterexample. Researchers have developed a range of verification techniques, mostly for real-valued ReLU networks. They typically formulate the verification as a Satisfiability Modulo Theory (SMT) problem [50, 28, 34, 20] or a Mixed Integer Linear Programming (MILP) problem [37, 12, 21, 17, 57], which can be exponentially slow due to the NP-completeness of the verification problem[34]. Certification (i.e., sound but incomplete verification) improves scalability at the cost of completeness [60, 59, 25, 62, 46, 18, 40, 52]. They often explore the idea of over-approximation to simplify the computation, which causes them unable to prove or disprove the specification in certain cases. Many of the certification methods can be unified under a layer-wise convex relaxation framework, but there seems to be an inherent barrier to tight verification via the relaxation captured by such a framework [48].

Until recently, exact verification of DNNs was too computationally expensive to scale beyond a few hundred neurons. Tjeng et al. [57] present the first exact verification of adversarial robustness for convolutional neural networks (CNNs) on MNIST using an MILP formulation. The verification performance is further improved by deriving tighter bounds of hidden neurons used in the MILP formulation [53]. Training a network with more stable ReLU neurons also reduces verification complexity and enables evaluating CNN robustness on the more challenging CIFAR10 dataset [61]. However, extant exact verifiers do not attempt to soundly model the floating point arithmetic in any inference implementation to achieve a manageable computational complexity [34, 57]. The resulting discrepancy between the actual and analyzed inference algorithms enables constructing adversarial examples for networks that are claimed to be robust by an exact verifier [32].

Binarized Neural Networks (BNNs) [29] constrain their activations and weights to be binary. Binarization facilitates analysis because its combinatorial nature enables close interaction with logical reasoning, which allows a rich set of properties to be encoded in SAT formulas. Examples include queries on adversarial robustness, Trojan attacks, fairness, network equivalence, and model counting [43, 4]. Analysis techniques for BNNs include efficient encoding [51] and exploiting decomposability between neurons or layers [13, 35]. However, prior BNN verifiers have exhibited even worse scalability than real-valued network verifiers. These scalability challenges have, to date, prevented the evaluation of any meaningful robustness metric for BNNs [44].

Adversarial attack and defense of DNNs is a developing field where most research focuses on real-valued networks [11, 2, 38, 58]. BNNs can be attacked by gradient-based adversaries [23] or

specialized solving algorithms [35]. We are the first to design robust training algorithms specifically for BNNs to induce verifiably robust networks.

## 3 Preliminaries

The Boolean Satisfiability Problem (SAT) is the problem of deciding whether there exists a variable assignment to satisfy a given Boolean expression [8]. We consider Boolean expressions in Conjunctive Normal Form (CNF) defined over a set of Boolean variables $x_1, \cdots, x_n$. A CNF $e$ is a conjunction of a set of *clauses*: $e = c_1 \wedge \cdots \wedge c_m$, where each clause $c_i$ is a disjunction of some *literals* $c_i = l_{i1} \vee \cdots \vee l_{is_i}$, and a literal $l_{ij}$ is either a variable or its negation: $l_{ij} = x_k$ or $l_{ij} = \neg x_k$. Despite the well known fact that 3-SAT is NP-complete [14], efficient heuristics have been developed to scale SAT solvers to handle industrial level problems [5].

Binarized Neural Networks (BNNs) [29] quantize their weights and activations to be binary. The BNN is proposed as a method to reduce the computation burden and speed up inference and possibly also training [47, 63, 31].

The basic building block of a BNN is a *linear-BatchNorm-binarize* module that maps an input tensor $x \in \{0, 1\}^n$ to an output tensor $y \in \{0, 1\}^m$ with a weight parameter $W \in \mathbb{R}^{m \times n}$ and also trainable parameters $\gamma \in \mathbb{R}^m$ and $\beta \in \mathbb{R}^m$ in the batch normalization [30]:

$$y = \text{bin}_{act}(\text{BatchNorm}(\text{bin}_w(W)x)) \tag{1}$$

where:

$$\text{bin}_w(W) = \text{sign}(W) \in \{-1, 1\}^{m \times n}$$

$$\text{sign}(x) = \begin{cases} 1 & \text{if } x \geq 0 \\ -1 & \text{otherwise} \end{cases}$$

$$\text{BatchNorm}(x) = \gamma \odot \frac{x - \text{E}[x]}{\sqrt{\text{Var}[x] + \epsilon}} + \beta$$

$$\text{where } \epsilon = 1\text{e}{-}5, \text{ and } \odot \text{ denotes element-wise multiplication}$$

$$\text{bin}_{act}(x) = (x \geq 0) = (\text{sign}(x) + 1)/2 \in \{0, 1\}^m$$

Note that we use $\{0, 1\}$ for activations rather than $\{-1, 1\}$ that is commonly adopted in the literature [29, 43]. Although both representations have identical representation capability because they are linear transformations of each other, our representation simplifies both the SAT encoding process and the zero padding of convolutional layers. The weight $\text{bin}_w(W)$ can take zero values in a sparse network.

**First layer:** The first layer of a BNN is usually applied on floating point inputs or fixed-point numbers [29]. However, encoding floating-point or integer arithmetic in SAT typically incurs high complexity. To simplify the verification process, we quantize the inputs:

$$x^{\text{q}} = \left\lfloor \frac{x}{s} \right\rceil \cdot s \tag{2}$$

where $x \in \mathbb{R}^n_{[0,1]}$ is the real-valued input, $x^{\text{q}}$ is the quantized input to be fed into the BNN, and $s$ is the quantization step size which can be set to $s = 1/255$ for emulating 8-bit fixed point values, or $2\epsilon$ for adversarial training with a $\ell_\infty$ perturbation bound of $\epsilon$. Since a robust network should be invariant to perturbations within $[x - \epsilon, x + \epsilon]$, we expect the quantization with $s = 2\epsilon$ not to discard information useful for robust classification, which is confirmed by checking that a few choices of the quantization step do not noticeably affect test accuracy.

**Last layer:** We consider the layer before softmax as the last layer of the BNN, which outputs a logits vector that can be interpreted as the classification score. We remove the $\text{bin}_{act}$ in (1) to obtain a real-valued score. To facilitate the SAT conversion, we also restrict the variance statistics and the scale parameter $\gamma$ in $\text{BatchNorm}(\cdot)$ of the last layer to be scalars computed on the whole feature map rather than per-channel statistics. In practice, such a restriction on the final batch normalization does not affect network performance. For the undefended `conv-small` network trained on CIFAR10, using the original per-channel batch normalization in the last layer achieves $54.74\%$ accuracy and a negative log-likelihood loss of $1.29$ on the test set, while the restricted network delivers $55.22\%$ test accuracy and the same loss. However, if the batch normalization is removed from the last layer, the test accuracy drops to $52.04\%$ and the loss increases to $1.35$.

# 4 Combinatorial analysis of BNNs

## 4.1 Encoding BNNs with reified cardinality constraints

We outline the encoding of a BNN and its robustness specification with respect to $\ell_\infty$-norm bounded perturbations around a given input as a set of Boolean clauses. The details are presented in the supplementary material.

Let $W^{\text{bin}} = \text{bin}_w(W)$ denote the sparse weights. The inference computation of a linear-BatchNorm-binarize module described in Section 3 can be formulated as:

$$y = \text{bin}_{act}(k^{\text{BN}} \odot (W^{\text{bin}}x) + b^{\text{BN}}) \tag{3}$$

where $k^{\text{BN}}$ and $b^{\text{BN}}$ are constants derived from batch normalization weights and statistics. Since values of $W^{\text{bin}}$ fall in $\{-1, 0, 1\}$, we can rewrite (3) as a set of *reified cardinality constraints* between $x$ and $y_i$:

$$y_i = \left( \sum_{j=1}^{n} l_{ij}(x_j) \gtreqqless \left[ b_i(k^{\text{BN}}, W^{\text{bin}}, b^{\text{BN}}) \right] \right) \tag{4}$$

where $l_{ij}(x)$ takes one value of $\{0, x, \neg x\}$ depending on $W_{ij}^{\text{bin}}$, $b(k^{\text{BN}}, W^{\text{bin}}, b^{\text{BN}})$ is a constant derived from the weights, $[\cdot]$ acts as flooring or ceiling according to the sign of $k^{\text{BN}}$, and $\gtreqqless$ acts as $\geq$ or $\leq$ according to the sign of the $k^{\text{BN}}$. In the encoding we treat TRUE and 1 equivalently, FALSE and 0 equivalently, and the comparator $\gtreqqless$ produces a value in $\{0, 1\}$.

Let $x$ denote the input image, $y$ denote the logits vector of the last layer, and $c$ denote the target class number. To evaluate robustness under $\ell_\infty$-norm bounded perturbations, we need to encode the input perturbation space $\|x - x_0\|_\infty \leq \epsilon$ and the untargeted attack goal $\vee_{i \neq c} (y_i - y_c > 0)$. Recall that inputs are quantized by $x^{\text{q}} = \left\lfloor \frac{x}{s} \right\rceil \cdot s$, which enables us to only encode the integer interval of allowed $\left\lfloor \frac{x}{s} \right\rceil$ values by merging the multiplier $s$ into $k^{\text{BN}}$ of the first layer. Encoding the constraint $v = \left\lfloor \frac{x}{s} \right\rceil \in \mathbb{Z} \cap [a, b]$ is achieved by introducing $b - a$ auxiliary Boolean variables $\{t_1, \cdots, t_{b-a}\}$ and assigning $v = a + \sum_{i=1}^{b-a} t_i$. We further reduce the search space by enforcing the thermometer encoding [9] on $\{t_1, \cdots, t_{b-a}\}$ so that the encoding of each value of $v$ is unique, via adding clauses $t_i \vee \neg t_j$ for $1 \leq i < j \leq b - a$. The clause $y_i - y_c > 0$ in the untargeted attack goal is converted into a reified cardinality constraint similarly to the layer encoding of (3) and (4).

Although the BNN is a special case of real-valued NNs, we still need to develop BNN-specific encodings because existing verifiers for real-valued NNs are unable to handle the $\text{sign}(\cdot)$ nonlinearity. It has also been shown that a straightforward MILP or ILP encoding for a BNN results in slower solving compared to the SAT encoding [43]. Note that computing the SAT encodings of BNNs only requires a linear scan of the network weights and architecture, and a verifier can cache the encoded logical constraints of the same network to be used for different inputs. Computing BNN encodings is easier and faster compared to computing the MILP encodings for real-valued networks that usually need to estimate the bounds of hidden neurons for each network input.

Compared to the SAT encoding in previous BNN verification research [43, 44], our network design and the corresponding encoding enjoy three benefits: (i) By replacing $\{-1, 1\}$ activations with $\{0, 1\}$ activations that directly correspond to Boolean values, the encoding is simplified and zero padding in convolutional layers is trivially supported, while Narodytska et al. [44] only evaluate fully connected networks; (ii) More information in the input is retained by quantization rather than binarization; and (iii) Test accuracy is improved by incorporating batch normalization in the last layer.

## 4.2 MiniSatCS: an efficient SAT solver for reified cardinality constraints

Modern SAT solvers typically build on the Conflict-Driven Clause Learning (CDCL) algorithm [39] to search for a solution of a set of disjunctive clauses. The strategy is to reduce the search space by learning new clauses from conflicts. There are three key procedures in this algorithm:

1. *Branching:* Pick an undecided variable and assign a value to it. The order of branching is usually decided by heuristics like VSIDS [41].

2. *Propagation:* Unit clauses are detected to infer undecided variables given the current variable assignment. A *unit clause* contains only one unassigned literal: if there is a clause $c = l_1 \vee \cdots \vee l_n$ in the database and $l_1, \ldots, l_{n-1}$ all evaluate to `FALSE` under current assignment, then $l_n$ must be `TRUE` so the whole clause could be satisfied.

3. *Clause learning:* When a clause evaluates to `FALSE` after a propagation step, the algorithm tracks the history of propagations that leads to the conflict, and determines the set of branching choices $x_1 = v_1, \cdots, x_k = v_k$ that are ultimately responsible for this conflict, where $x_i$ is a branching variable and $v_i \in \{$`TRUE`, `FALSE`$\}$ is the branching decision. A new clause $(x_1 = \neg v_1) \vee \cdots \vee (x_k = \neg v_k)$ is inserted into the database of learned clauses to facilitate future propagation.

We extend a CDCL-based SAT solver to natively handle reified cardinality constraints. We compare our technique with two existing approaches for solving a Boolean system with such constraints:

- Deploy an encoding algorithm to convert each reified cardinality constraint into a set of disjunctive clauses to be solved by an off-the-shelf SAT solver. The sequential counters encoder [54], which needs $O(nb)$ auxiliary variables and clauses for a constraint with $n$ literals and a bound of $b$, has been used by previous BNN verifiers [43, 44]. An arguably more efficient encoder, the cardinality networks [1] that need $O(n \log^2 b)$ auxiliary variables and clauses, has been adopted for learning transition models from BNNs for planning [49].

- Solve the pseudo-Boolean constraints derived from the original constraints. Pseudo-Boolean constraints allow Boolean literals to be multiplied by integer weights. A reified cardinality constraint $y = (\sum_{i=1}^{n} l_i \geq b)$ can be encoded by two linear pseudo-Boolean constraints:

$$\begin{cases} \sum_{i=1}^{n} l_i + b \cdot \neg y \geq b & \text{which encodes } \sum_{i=1}^{n} l_i < b \implies y = 0 \\ (n - b + 1) \cdot y - 1 \geq \sum_{i=1}^{n} l_i - b & \text{which encodes } \sum_{i=1}^{n} l_i \geq b \implies y = 1 \end{cases} \quad (5)$$

  We are unaware of previous use of this encoding in BNN analysis. We include this comparison as one potential alternative to our direct support of reified cardinality constraints.

Our extension builds on the observation that the CDCL framework can be generalized to handle clauses not in the disjunctive form, as long as each clause permits inferring values of undecided variables. This idea has been explored in the literature to extend SAT solvers to domain-specific problems [55, 36, 24]. A key component of our extension is to efficiently handle the propagation through a reified cardinality constraint $y = (\sum_{i=1}^{n} l_i \leq b)$, which contains two cases:

- *Operand-inferring:* If $y$ is known and enough of the $\{l_i\}$ are known, then the remaining $\{l_i\}$ can be inferred. For example, if $y$ is known to be true and there are already $b$ literals in $\{l_i\}$ known to be true, then the other literals must be false.

- *Target-inferring:* If enough of the $\{l_i\}$ are known, then $y$ can be inferred. For example, if the number of false literals in $\{l_i\}$ reaches $n - b$, then $y$ can be inferred to be true.

We present `MiniSatCS`, a novel system that efficiently and natively handles reified cardinality constraints based on the above ideas. `MiniSatCS` extends `MiniSat 2.2` which is a highly optimized and minimalistic SAT solver based on years of SAT solving research [19]. We extend the clause data structure to represent both the reified less-equal cardinality constraints and disjunction constraints. `MiniSatCS` maintains counters that keep the current number of known true or false literals for each clause. The counters are updated when related variable assignment changes by a notification system similar to watched literals [41], so repeated scanning of clauses is avoided during search for unit clauses. We use random polarity and turned off the phase saving heuristics [45] in the solver since it is faster for BNN verification.

We compare the performance of `MiniSatCS` on the `MNIST-MLP` network (defined in Section 7) against `MiniSat 2.2` with two encoding choices, two other state-of-the-art solvers, and prior BNN verification: (i) `MiniSat-seqcnt` that applies `MiniSat 2.2` on the sequential counters encoding [54]; (ii) `MiniSat-cardnet` that applies `MiniSat 2.2` on the cardinality networks encoding [1]; (iii) a general SMT solver `Z3` [15] that supports pseudo-Boolean logic via on-demand compilation into sorting circuits; (iv) `RoundingSat` [16] with the encoding outlined in (5), which is a state-of-the-art pseudo-Boolean solver accelerated by incremental linear program solving, with native handling of linear pseudo-Boolean constraints; and (v) the end-to-end solving time for verifying the same BNN

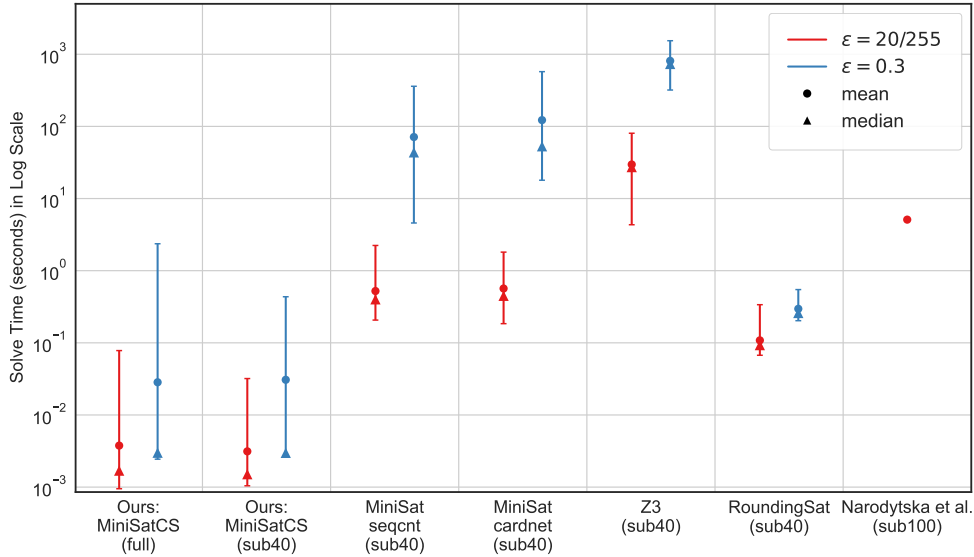

Figure 1: Performance comparison on searching adversarial inputs for an undefended `MNIST-MLP` network. The graph plots the min/median/mean/max times of each solver on the indicated set of benchmarks, with a one hour time limit. The (`sub40`) or (`sub100`) suffix indicates testing on a subset containing 40 or 100 MNIST test images, and (`full`) indicates testing on the complete test set. Note that the solving time is in log scale, and our method exhibits especially low median solving times. Our `MNIST-MLP` network achieves 97.14% test accuracy with 18K non-zero parameters and an input quantization step $s = 0.1$, compared to 95.2% accuracy with 20K non-zero parameters and binarized inputs reported by Narodytska et al. [44].

architecture reported in prior state-of-the-art BNN verification resarch [44], which uses the Glucose SAT solver [3] with the sequential counters encoding. We present the solving time details in Figure 1. Our results show that `MiniSatCS` is several orders of magnitude faster than other solvers and the prior state-of-the-art result of BNN verification on the same architecture. We present more results in the supplementary material.

## 5  Training solver-friendly BNNs

### 5.1  BinMask: balanced weight sparsifying

Sparse weights are known to facilitate DNN verification [57, 61, 44]. BNNs are commonly sparsified by ternary quantization [44] that sets $\text{bin}_w(W) = 0$ when $|W| < T$. However, ternary quantization suffers from two drawbacks: (i) The threshold $T$ and the penalty coefficient for $\ell_1$ regularization are two coupled parameters that need tuning; and (ii) For convolutional networks, the sparsity of convolutional layers is usually lower than that of fully connected layers, which has also been observed during pruning real-valued networks [27]. The low sparsity of the convolutional layers limits verification speedup because those layers reuse the weights and lower weight sparsity leads to higher verification complexity. While it is possible to prune each layer with a fixed rate and retrain the network iteratively [22], such methods are especially costly for adversarial training.

We propose to induce more balanced layer-wise sparsity by decoupling the optimization of weight value and weight sparsity. Our method also eliminates the threshold hyperparameter. A binary mask is applied on the weights, so that in the training process, the weights become gradually more sparse by masking out less important weights and the sign of unmasked weights can still be optimized. More formally, for each weight $W$ we introduce a new mask weight $M_W$ to be optimized independently of $W$ and replace the definition of $\text{bin}_w$ in (1) by:

$$\text{bin}_w(W) = \text{sign}(W) \odot \frac{\text{sign}(M_W) + 1}{2} \qquad (6)$$

Table 1: Comparing BinMask and ternary quantization on undefended `conv-small` networks. While both methods produce similar total sparsity, the more balanced layer-wise sparsity of BinMask results in faster verification. Total sparsity is the proportion of zero parameters in the whole network, which is largely determined by the sparsity of the third layer — a fully connected layer with a large weight matrix. Solve time is measured by applying `MiniSatCS` on $40$ randomly chosen test images with a one hour time limit.

| | MNIST $\epsilon = 0.1$ | | CIFAR10 $\epsilon = 2/255$ | |
| --- | --- | --- | --- | --- |
| Sparsifier | Ternary | BinMask | Ternary | BinMask |
| Total Sparsity | 81% | 84% | 82% | 79% |
| Layer-wise Sparsity | 16% 40% 84% 36% | 91% 90% 83% 92% | 16% 48% 84% 23% | 84% 85% 79% 87% |
| Mean Solve Time (sub40) | 756.288 | 0.002 | 0.267 | 0.003 |
| Max Solve Time (sub40) | 3600.001 | 0.007 | 5.707 | 0.005 |
| Natural Test Accuracy | 97.59% | 97.35% | 54.78% | 55.22% |

Table 2: Effect of Cardinality Bound Decay (CBD) on adversarially trained `conv-large` networks. The CBD loss effectively reduces cardinality bounds, resulting in significant verification speedup. The results suggest that it also improves robustness, perhaps by regularizing model capacity. Solve time is measured by applying `MiniSatCS` on $40$ randomly chosen test images with a one hour time limit. Verifiable accuracy is evaluated on the complete test set without time limit for networks that can be verified within one second per case on average.

| | MNIST $\epsilon = 0.3$ | | | | CIFAR10 $\epsilon = 8/255$ | | | |
| --- | --- | --- | --- | --- | --- | --- | --- | --- |
| CBD Loss Penalty ($\eta$) | 0 | 1e−5 | 1e−4 | 5e−4 | 0 | 1e−5 | 1e−4 | 5e−4 |
| Mean / Max Card Bound | 148.3 / 364.0 | 4.3 / 15.1 | 3.2 / 9.0 | 2.7 / 6.8 | 123.8 / 312.0 | 3.1 / 7.4 | 2.7 / 5.9 | 2.1 / 6.4 |
| Mean Solve Time (sub40) | 2442.698 | 32.776 | 0.009 | 0.005 | 206.037 | 0.010 | 0.009 | 0.009 |
| Max Solve Time (sub40) | 3600.006 | 1287.739 | 0.040 | 0.012 | 3600.001 | 0.019 | 0.013 | 0.014 |
| Verifiable Accuracy | - | - | 69.04% | 72.48% | - | 19.28% | 18.81% | 20.08% |
| Natural Test Accuracy | 98.88% | 97.37% | 96.97% | 96.26% | 53.91% | 40.80% | 38.75% | 35.17% |
| PGD Accuracy | 89.23% | 87.60% | 87.82% | 87.82% | 15.32% | 27.06% | 26.22% | 24.69% |
| First Layer / Total Sparsity | 85% / 82% | 86% / 89% | 83% / 89% | 82% / 87% | 95% / 88% | 95% / 94% | 94% / 87% | 89% / 90% |

The mask $M_W$ is initialized from a folded normal distribution (i.e., absolute value of Gaussian) so that the training starts from a dense network. Weight decay is applied on the positive elements in the mask $M_W$. We call our formulation *BinMask* and present results of an empirical comparison with ternary quantization in Table 1. Although the two methods achieve similar total sparsity and test accuracy, BinMask induces more balanced layer-wise sparsity, and consequently its verification is thousands of times faster. We present additional experiments and further discussions in the supplementary material.

## 5.2 Cardinality bound decay

While BinMask alone sparsifies the small network enough to be efficiently verified, it is not sufficient for a larger network. To further reduce verification complexity, we revisit the reified cardinality constraint $y = (\sum_{i=1}^{n} l_i \leq b)$ and note the following facts:

1. The sequential counters encoder [54] that converts the constraint into CNF needs $O(nb)$ auxiliary variables and clauses. Thus smaller $b$ produces a simpler encoding.

2. `MiniSatCS` can infer $y$ to be false once the number of true literals in $\{l_i\}$ exceeds $b$, and a smaller $b$ increases the likelihood of this inference.

3. If the literals $\{l_i\}$ are drawn from independent symmetrical Bernoulli distributions, then the entropy of $y$ is a symmetrical concave function with respect to $b$ which is maximized when $b = \frac{n}{2}$. Therefore the further $b$ deviates from $\frac{n}{2}$, the more predictable $y$ becomes.

We are thus motivated to regularize the bound in reified cardinality constraints to reduce verification complexity. We propose a Cardinality Bound Decay (CBD) loss to achieve this goal, by adding an $\ell_1$ penalty on the bias terms in (4) that exceed a threshold $\tau$:

$$L^{\mathrm{CBD}} = \eta \max \left( b(k^{\mathrm{BN}}, W^{\mathrm{bin}}, b^{\mathrm{BN}}) - \tau, 0 \right) \qquad (7)$$

where $\eta$ is a coefficient controlling the strength of this loss. We set $\tau = 5$ in our experiments because it is an empirically good choice among a few values that enables fine control of the accuracy-speed

tradeoff by tuning $\eta$. Note that since $\sum_{i=1}^{n} l_i \leq b$ is equivalent to $\sum_{i=1}^{n} \neg l_i \geq n - b$, we only need to minimize $b$ rather than maximize $\left| b - \frac{n}{2} \right|$ in this loss. The removal of $n$ results in a simpler formulation. We present an empirical evaluation of the CBD loss in Table 2, which shows that the CBD loss reduces the cardinality bounds and verification complexity significantly. Interestingly, the results also suggest that the CBD loss serves as an effective regularizer which favors model robustness over natural test accuracy. Note that the cardinality bound is another measurement of verification complexity different from sparsity, since they are not strongly correlated.

## 6 Training robust BNNs with adaptive gradient cancelling

Although the sign function has zero gradient almost everywhere, a BNN can still be trained using gradient based optimizers by adopting the straight-through estimator [6] which treats the sign function as an identity function during backpropagation. This approach also enables training robust BNNs with a projected gradient descent (PGD) adversary, similar to training robust real-valued networks [38].

For a sign function quantization $q = \text{sign}(r)$, empirical results suggest that gradient cancelling via backpropagating with $g_r = g_q \mathbb{1}_{|r| \leq 1}$ improves both training and test accuracy [29], where $g_q$ and $g_r$ are the gradients of the loss with respect to $q$ and $r$ respectively. Gradient cancelling can be seen as computing the gradient as if the forward propagation is hard tanh $q = \text{htanh}(r) = \text{clip}(r, -1, 1)$.

Using the hard tanh gradient cancelling in the PGD adversarial training produces networks with a high PGD test accuracy, but the true adversarial accuracy evaluated by an exact verifier still remains low. This suggests that the PGD attack becomes ineffective. To improve the robustness, we instead propose to use standard tanh (i.e., backpropagating with $g_r = (1 - \tanh^2 r) g_q$) to provide more effective gradients for both training and the PGD attack. Using tanh gradient cancelling in PGD improves the attack success rate on a `conv-small` network from 19.82% to 23.25%.

We generalize the idea and propose adaptive gradient cancelling: $g_r = (1 - \tanh^2(\alpha r)) g_q$, where $\alpha$ is a global parameter to control the strength of gradient cancelling. Motivated by the intuition that a stronger PGD attack indicates better gradients, $\alpha$ is chosen to maximize PGD attack success rate at the beginning of each epoch. The robustness of the `conv-large` networks is further improved by generating adversarial examples with the `MiniSatCS` verifier for images where the PGD attack fails in the last ten epochs. Table 3 compares different gradient computing methods.

Table 3: Comparing gradient computing methods for adversarial training on CIFAR10 with the `conv-large` network and $\epsilon = 8/255$. The PGD accuracy is evaluated on the test set with the same gradient computing as in training. The verifiable accuracy is the exact adversarial robustness. Tanh gradient cancelling improves all the metrics, and adaptive gradient cancelling reduces the gap between PGD accuracy and verifiable accuracy.

|  | hard tanh | tanh | adaptive | adaptive + verifier adv |
|---|---|---|---|---|
| Natural Test Accuracy | 35.42% | 38.79% | 35.17% | 35.00% |
| PGD Accuracy | 22.79% | 24.98% | 24.69% | 26.41% |
| Verifiable Accuracy | 11.13% | 14.70% | 20.08% | 22.55% |

## 7 Experiments

We adopt three network architectures from the literature for the evaluation of EEV. `MNIST-MLP` is a binarized multilayer perceptron with hidden layers having $[500, 300, 200, 100, 10]$ units [44]. `Conv-small` is a network with two convolutional layers of 16 and 32 channels, followed by two fully connected layers with 100 and 10 units. The convolutional layers have $4 \times 4$ filters and $2 \times 2$ stride with a padding of 1. `Conv-large` has four convolutional layers with $[32, 32, 64, 64]$ channels and $[3 \times 3, 4 \times 4, 3 \times 3, 4 \times 4]$ spatial filters respectively, where each layer has a padding of 1 and the larger convolutions have a stride of $2 \times 2$. They are followed by three fully connected layers with $[512, 512, 10]$ output units. The convolutional BNNs have the same architectures as in [61] except that we binarize the network. We present details of the experimental settings in the supplementary material. Our source code is available at `https://github.com/jia-kai/eevbnn`.

**Verifying adversarial robustness:** We evaluate the performance of EEV on the task of verifying robustness of BNNs against $\ell_\infty$-bounded input perturbations on the MNIST and CIFAR10

Table 4: Results of verifying adversarial robustness. EEV verifies BNNs significantly faster with comparable verfiable accuracy in most cases.

| | | Mean Time (s) | | | Accuracy | | | Timeout |
|---|---|---|---|---|---|---|---|---|
| | | Build | Solve | Total | Verifiable | Natural | PGD | |
| MNIST $\epsilon = 0.1$ | EEV S | 0.0158 | 0.0004 | 0.0162 | 89.29% | 97.44% | 93.47% | 0 |
| | EEV L | 0.1090 | 0.0025 | 0.1115 | 91.68% | 97.46% | 95.47% | 0 |
| | Xiao et al. [61] S | 4.98 | 0.49 | 5.47 | 94.33% | 98.68% | 95.13% | 0.05% |
| | Xiao et al. [61] L[*] | 156.74 | 0.27 | 157.01 | 95.6% | 98.95% | 96.58% | 0 |
| MNIST $\epsilon = 0.3$ | EEV S | 0.0140 | 0.0006 | 0.0146 | 66.42% | 94.31% | 80.70% | 0 |
| | EEV L | 0.1140 | 0.0039 | 0.1179 | 77.59% | 96.36% | 87.90% | 0 |
| | Xiao et al. [61] S | 4.34 | 2.78 | 7.12 | 80.68% | 97.33% | 92.05% | 1.02% |
| | Xiao et al. [61] L[*] | 166.39 | 37.45 | 203.84 | 59.6% | 97.54% | 93.25% | 24.1% |
| CIFAR10 $\epsilon = \frac{2}{255}$ | EEV S | 0.0258 | 0.0013 | 0.0271 | 26.13% | 46.58% | 33.70% | 0 |
| | EEV L | 0.1653 | 0.0097 | 0.1750 | 30.49% | 47.35% | 38.22% | 0 |
| | Xiao et al. [61] S | 52.58 | 13.50 | 66.08 | 45.93% | 61.12% | 49.92% | 1.86% |
| | Xiao et al. [61] L[*] | 335.97 | 29.88 | 365.85 | 41.4% | 61.41% | 50.61% | 9.6% |
| CIFAR10 $\epsilon = \frac{8}{255}$ | EEV S | 0.0313 | 0.0014 | 0.0327 | 18.93% | 37.75% | 24.60% | 0 |
| | EEV L | 0.1691 | 0.0090 | 0.1781 | 22.55% | 35.00% | 26.41% | 0 |
| | Xiao et al. [61] S | 38.34 | 22.33 | 60.67 | 20.27% | 40.45% | 26.78% | 2.47% |
| | Xiao et al. [61] L[*] | 401.72 | 20.14 | 421.86 | 19.8% | 42.81% | 28.69% | 5.4% |

EEV is exact verification for BNNs with the proposed EEV system. Xiao et al. [61] is exact verification for real-valued networks with data taken from [61]. The S and L suffix indicates `conv-small` or `conv-large` architectures. The build time is the time required to generate the SAT or MILP formulation from the network weights and the input image. The solve time is the time that the solver spends solving the formulation. We limit the solving time for each case to 120 seconds as in [61].

[*] Xiao et al. [61] test their large model only on the first 1000 images due to slow verification.

Table 5: Verifying a model ensemble with reject option on MNIST with a $\ell_\infty$ perturbation bound of 0.3

| | Test Accuracy | Mean Solve Time (s) | Attack Success Rate |
|---|---|---|---|
| `conv-small` | 94.31% | 0.001 | 33.58% |
| `conv-large` | 96.36% | 0.004 | 22.41% |
| ensemble | 93.32% | 0.003 | 17.21% |

benchmarks. The networks are trained with adaptive gradient cancelling. We present our results in Table 4. It shows that EEV achieves solving times 109.29 to 15897.42 times faster (and total times 338.28 to 2440.65 times faster) than state-of-the-art exact verification of real-valued networks [61] with comparable verifiable accuracy in most cases.

**Verifying a model ensemble with reject option:** We evaluate the extensibility of our system by considering an ensemble of $M$ models that rejects the input if they do not fully agree on the classification. A successful adversarial attack must present an input that causes all components to produce the same wrong classification. The attack goal can be easily formulated in CNF. Let $n$ be the number of classes, $c$ be the correct class, and $r_{ij}^m = (y_i^m - y_j^m > 0)$ being a reified cardinality constraint denote whether the score of class $i$ is higher than that of class $j$ in the logits vector of model $m$. Let $f_i = \wedge_{1 \le m \le M} \wedge_{1 \le j \ne i \le n} r_{ij}^m$ denote whether all models agree on class $i$. Then the attack goal is simply $\vee_{1 \le i \ne c \le n} f_i$. We present in Table 5 the results of verifying an ensemble of two models. The results show that EEV can easily handle this complex query (i.e., $\arg\max f_1(x) = \arg\max f_2(x) \ne c$) which is arguably nontrivial or even challenging to be formulated as an efficiently solvable continuous optimization problem [33], and that our solver is efficient in exploring the decision space of a model ensemble.

# 8 Conclusion

In this work we demonstrate that it is possible to significantly scale up the exact verification of Binarized Neural Networks (BNNs) by equipping an off-the-shelf SAT solver with domain-specific propagation rules and simultaneously training solver-friendly robust BNNs. Although we focus on verifying adversarial robustness, our method could be generalized to verify other properties of BNNs. Our experimental results demonstrate the significant performance gains that our techniques deliver.

## Broader impact

Binarized Neural Networks (BNNs) are attractive targets for deployment in a variety of contexts including edge devices due to their efficiency advantages over real-valued networks. The present research, by developing techniques for training and verifying robust BNNs, may help enable the development of systems that more reliably and predictably serve their goals — systems that may be less likely to exhibit unexpected behavior in response to new inputs; systems that may be less vulnerable to attack. The research may therefore increase the range and capabilities of systems that use BNNs. Because this technology is general purpose and may be deployed in the service of prosocial, antisocial, or mixed goals, the ultimate broader impacts may be shaped by the choices societies make about how to use these capabilities. Example potential impacts include the increased deployment of accurate surveillance systems, more accurate vision systems for safer self-driving cars, more reliable autonomous control systems, and less effort spent certifying systems that include BNNs.

## Acknowledgments and Disclosure of Funding

This work was supported by the Grass Instrument Company Fellowship in Electrical Engineering and the project "Automatically Learning the Behavior of Computational Agents" (MIT CO 6940111, sponsored by Boeing with sponsor ID #Z0918-5060). We thank all the reviewers for providing the insightful comments that help further improve the quality of this paper, especially during this difficult year. Kai would like to thank Vijay Ganesh and Saeed Nejati for the helpful discussions on SAT solving, and Dimitris Tsipras for discussing training robust neural network. Kai also thanks Qi Song for her constant support during working on this project in the quarantine.

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
