[Supplementary Material]

# Efficient Exact Verification of Binarized Neural Networks (Supplementary Material)

**Kai Jia**
MIT CSAIL
32 Vassar St, Cambridge, MA 02139
jiakai@mit.edu

**Martin Rinard**
MIT CSAIL
32 Vassar St, Cambridge, MA 02139
rinard@csail.mit.edu

## 1 Details of BNN encoding

We present details for encoding the inference computation of a single linear-BatchNorm-binarize module in a BNN. Recall that such a module is defined for an input $x \in \{0, 1\}^n$, an output $y \in \{0, 1\}^m$, and a weight $W \in \mathbb{R}^{m \times n}$:

$$y = \text{bin}_{act}(\text{BatchNorm}(\text{bin}_w(W)x)) \tag{1}$$

where:

$$\text{bin}_w(W) = \text{sign}(W) \in \{-1, 1\}^{m \times n}$$

$$\text{BatchNorm}(x) = \gamma \odot \frac{x - \text{E}[x]}{\sqrt{\text{Var}[x] + \epsilon}} + \beta$$

is the Batch Normalization [1] with parameters $\gamma \in \mathbb{R}^m$ and $\beta \in \mathbb{R}^m$

$$\text{bin}_{act}(x) = (x \geq 0) = (\text{sign}(x) + 1)/2 \in \{0, 1\}^m$$

Batch Normalization becomes a linear transformation in inference:

$$x^{\text{BN}} = k^{\text{BN}} \odot x + b^{\text{BN}} \tag{2}$$

where:

$$k^{\text{BN}} = \frac{\gamma}{\sqrt{\sigma^2 + \epsilon}}$$

$$b^{\text{BN}} = \beta - k^{\text{BN}}\mu$$

$\mu$ is the mean of $x$ on training set

$\sigma^2$ is the variance of $x$ on training set

With $W^{\text{bin}} = \text{bin}_w(W)$ being a fixed parameter, we can rewrite the computation of a single element of $y$ in (1) as the following:

$$y_i = \left( k_i^{\text{BN}} \sum_{j=1}^n W_{ij}^{\text{bin}} x_j + b_i^{\text{BN}} \geq 0 \right) \tag{3}$$

To encode (3) as a reified cardinality constraint, we consider $1$ and TRUE interchangeably and $0$ and FALSE interchangeably. If $W_{ij}^{\text{bin}} = 1$, we have $W_{ij}^{\text{bin}}x_j = x_j$, and if $W_{ij}^{\text{bin}} = -1$, we rewrite

$W_{ij}^{\text{bin}}x_j = -x_j = (1 - x_j) - 1 = \neg x_j - 1$. With such substitutions of $W_{ij}^{\text{bin}}x_j$ we obtain a reified cardinality constraint:

$$
\begin{aligned}
y_i &= \left( k_i^{\text{BN}} \sum_{j=1}^{n} W_{ij}^{\text{bin}} x_j + b_i^{\text{BN}} \geq 0 \right) \\
&= \left( k_i^{\text{BN}} \left( \sum_{j=1}^{n} l_{ij}(x_j) + b_i^{\text{SAT}} \right) + b_i^{\text{BN}} \geq 0 \right) \\
&= \left( \sum_{j=1}^{n} l_{ij}(x_j) \gtreqless \left[ b_i(k^{\text{BN}}, W^{\text{bin}}, b^{\text{BN}}) \right] \right)
\end{aligned}
\tag{4}
$$

where:

$$
l_{ij}(x_j) = \begin{cases} x_j & \text{if } W_{ij}^{\text{bin}} = 1 \\ 0 & \text{if } W_{ij}^{\text{bin}} = 0 \\ \neg x_j & \text{if } W_{ij}^{\text{bin}} = -1 \end{cases}
$$

$$
b_i^{\text{SAT}} = \sum_{j=1}^{n} \min\left( W_{ij}^{\text{bin}}, 0 \right)
$$

$$
b_i(k^{\text{BN}}, W^{\text{bin}}, b^{\text{BN}}) = -\frac{b_i^{\text{BN}}}{k_i^{\text{BN}}} - b_i^{\text{SAT}}
$$

$$
\left( x \gtreqless y \right) = \begin{cases} x \geq y & \text{if } k_i^{\text{BN}} > 0 \\ x \leq y & \text{if } k_i^{\text{BN}} < 0 \end{cases}
$$

$$
[x] = \begin{cases} \lceil x \rceil & \text{if } k_i^{\text{BN}} > 0 \\ \lfloor x \rfloor & \text{if } k_i^{\text{BN}} < 0 \end{cases}
$$

## 2   Experimental details

**Experimental environment**   We conduct our experiments on a workstation equipped with two GPUs (NVIDIA Titan RTX and NVIDIA GeForce RTX 2070 SUPER), 128 GiB of RAM and an AMD Ryzen Threadripper 2970WX 24-core processor. We use the PyTorch [3] framework to train all the networks.

**Training method**   We train the networks using the Adam optimizer [2] for 200 epochs with a minibatch size of 128, with the exception of the undefended `conv-small` networks on CIFAR10 which is trained for only 90 epochs to avoid overfitting. Due to fluctuations of test accuracy between epochs, we select from the last three epochs the model with the highest natural test accuracy or PGD accuracy on the first 40 training minibatches. The mean and variance statistics of batch normalization layers are recomputed on the whole training set after training finishes. Learning rate is initially $1\mathrm{e}{-}4$ and decayed to half on epoch 150. We use PGD with adaptive gradient cancelling to train robust networks, where the perturbation bound $\epsilon$ is increased linearly from 0 to the desired value in the first 100 epochs and the number of PGD iteration steps grows linearly from 0 to 10 in the first 50 epochs.

The parameter $\alpha$ in adaptive gradient cancelling is chosen to maximize the PGD attack success rate evaluated on 40 minibatches of training data sampled at the first epoch. Candidate values of $\alpha$ are between 0.6 to 3.0 with a step of 0.4. Note that $\alpha$ is a global parameter shared by all neurons.

We do not use any data augmentation techniques for training. Due to limited computing resource and significant differences between the settings we considered, data in this paper are reported based on one evaluation run.

**Weight initialization**   All weights for the convolutional or fully connected layers are initialized from a Gaussian distribution with standard deviation 0.01, and the mask weights $M_W$ in BinMask are enforced to be positive by taking the absolute value during initialization.

Table 1: Verification time with varying perturbation bounds. Time limit is 120 seconds.

| Dataset | Network | Mean Solve Time | | | Solver Timeout | | | Mean Build+Solve Time | | |
|---------|---------|---------|---------|---------|---------|---------|---------|---------|---------|---------|
| Training $\epsilon$ | | $\epsilon = \epsilon_0$ | $\epsilon = \epsilon_1$ | $\epsilon = \epsilon_2$ | $\epsilon = \epsilon_0$ | $\epsilon = \epsilon_1$ | $\epsilon = \epsilon_2$ | $\epsilon = \epsilon_0$ | $\epsilon = \epsilon_1$ | $\epsilon = \epsilon_2$ |
| MNIST | `conv-small` | 0.0004 | 0.0021 | 0.0713 | 0 | 0 | 0.01% | 0.0162 | 0.0180 | 0.0876 |
| $\epsilon = 0.1$ | `conv-large` | 0.0025 | 0.0129 | 0.1269 | 0 | 0 | 0.01% | 0.1115 | 0.1197 | 0.2254 |
| MNIST | `conv-small` | 0.0004 | 0.0004 | 0.0006 | 0 | 0 | 0 | 0.0147 | 0.0155 | 0.0146 |
| $\epsilon = 0.3$ | `conv-large` | 0.0010 | 0.0018 | 0.0039 | 0 | 0 | 0 | 0.1162 | 0.1142 | 0.1179 |
| CIFAR10 | `conv-small` | 0.0013 | 0.0017 | 0.0025 | 0 | 0 | 0 | 0.0271 | 0.0298 | 0.0366 |
| $\epsilon = 2/255$ | `conv-large` | 0.0097 | 0.0136 | 0.0141 | 0 | 0 | 0 | 0.1750 | 0.1947 | 0.1918 |
| CIFAR10 | `conv-small` | 0.0009 | 0.0011 | 0.0014 | 0 | 0 | 0 | 0.0236 | 0.0284 | 0.0327 |
| $\epsilon = 8/255$ | `conv-large` | 0.0087 | 0.0084 | 0.0090 | 0 | 0 | 0 | 0.1704 | 0.1696 | 0.1781 |

**Other hyperparameters**  The input quantization step $s$ is set to be $0.61$ for training on the MNIST dataset, and $0.064 \approx 16.3/255$ for CIFAR10, which are chosen to be slightly greater than twice the largest perturbation bound we consider for each dataset. The CBD loss is applied on `conv-large` networks only and $\eta$ is set to be $5e{-}4$ unless otherwise stated. We apply a weight decay of $1e{-}7$ on the binarized mask weight $M_W$ of BinMask for `conv-small` and `conv-large` networks, and the weight decay is $1e{-}5$ for the `MNIST-MLP` network. PGD accuracies reported for the test set are evaluated with $100$ steps of PGD iterations.

## 3   Adversarial robustness against varying perturbation bounds

We run the verifier with varying perturbation bounds and present the time in Table 1 and the accuracy in Table 2. The bounds are set to be $\epsilon_0 = 0.1$, $\epsilon_1 = 0.2$, and $\epsilon_2 = 0.3$ for MNIST and $\epsilon_0 = 2/255$, $\epsilon_1 = 5/255$, and $\epsilon_2 = 8/255$ for CIFAR10. We note a few interesting discoveries:

1. For the same network, the verification becomes slower for larger perturbation bounds. This behavior is expected because a larger bound corresponds to a larger input perturbation space, which makes the verification problem harder to solve.

2. For the same bound used in verification, verification is faster for a more robust network that is trained with a larger perturbation bound. One plausible explanation is that robust networks are less sensitive to input changes, and they naturally allow the solver to learn simpler clauses that describe relationships between neurons. Another factor is that stronger adversarial training usually (but not always) induces more sparse weights as can be seen in Table 2.

3. The gap between PGD accuracy and verifiable accuracy for a fixed test perturbation bound gets narrower as the training perturbation becomes stronger. More interestingly, when tested against perturbations of $\epsilon_0$, although the network has higher verifiable accuracy when it is trained with stronger perturbations, its PGD accuracy even gets lower (comparing PGD accuracy with $\epsilon = \epsilon_0$ in Table 2). Such phenomenon suggests that PGD accuracy is not always positively correlated with verfiable accuracy and the adversarial training algorithm could be further improved.

## 4   Ablation study

We conduct comprehensive experiments to study the effectiveness of our proposed methods — namely BinMask, CBD loss, and native handling of the reified cardinality constraints in the `MiniSatCS` verifier — under different settings. We apply various combinations of ternary quantization, BinMask, and CBD loss during training, and verify the networks using multiple solvers. The experimental results are presented in Table 3 and Table 4.

For each dataset, we train the `conv-small` and `conv-large` networks under two training settings: undefended (i.e., $\epsilon = 0$) and adversarial training with a large perturbation bound ($\epsilon = 0.3$ for MNIST and $\epsilon = 8/255$ for CFAR10). We tune the weight decay coefficient of ternary quantization so that the total sparsity is close to that of BinMask or BinMask+CBD. All BinMask networks have the same

Table 2: Verifiable accuracy with varying perturbation bounds.

| Dataset Training $\epsilon$ | Network | Test Accuracy | PGD Adversarial Accuracy | | | Verifiable Accuracy | | | Sparsity |
|---|---|---|---|---|---|---|---|---|---|
| | | | $\epsilon = \epsilon_0$ | $\epsilon = \epsilon_1$ | $\epsilon = \epsilon_2$ | $\epsilon = \epsilon_0$ | $\epsilon = \epsilon_1$ | $\epsilon = \epsilon_2$ | |
| MNIST | conv-small | 97.44% | 93.47% | 86.22% | 70.68% | 89.29% | 66.49% | 25.45% | 90% |
| $\epsilon = 0.1$ | conv-large | 97.46% | 95.47% | 92.56% | 86.98% | 91.68% | 75.35% | 40.14% | 91% |
| MNIST | conv-small | 94.31% | 91.74% | 87.43% | 80.70% | 90.24% | 82.14% | 66.42% | 94% |
| $\epsilon = 0.3$ | conv-large | 96.36% | 94.82% | 92.19% | 87.90% | 93.71% | 88.55% | 77.59% | 87% |
| CIFAR10 | conv-small | 46.58% | 33.70% | 18.85% | 9.32% | 26.13% | 8.26% | 2.39% | 94% |
| $\epsilon = 2/255$ | conv-large | 47.35% | 38.22% | 28.20% | 19.60% | 30.49% | 13.30% | 4.98% | 85% |
| CIFAR10 | conv-small | 37.75% | 33.88% | 29.02% | 24.60% | 32.18% | 24.82% | 18.93% | 96% |
| $\epsilon = 8/255$ | conv-large | 35.00% | 32.45% | 29.17% | 26.41% | 31.20% | 26.39% | 22.55% | 98% |

Table 3: Comparison of methods on 40 randomly chosen MNIST test images with solving time limit of 3600 seconds.

| $\epsilon_{\text{train}}$ | Network Architecture | Training Method | Test Accuracy | Sparsity | Solver | Mean Solve Time | Median Solve Time | Timeout | Verifiable Accuracy |
|---|---|---|---|---|---|---|---|---|---|
| 0 | conv-small | Ternary | 97.59% | 81% | MiniSatCS | 756.288 | 4.281 | 15% | 0% |
| | | BinMask | 97.35% | 84% | MiniSatCS | 0.002 | 0.002 | 0 | 52% |
| | | | | | MiniSat | 2.249 | 1.142 | 0 | 52% |
| | | | | | Z3 | 0.089 | 0.089 | 0 | 52% |
| | | | | | RoundingSat | 0.048 | 0.042 | 0 | 52% |
| | conv-large | Ternary | 99.07% | 86% | MiniSatCS | 2522.082 | 3600.002 | 68% | 0% |
| | | Ternary+CBD | 95.58% | 87% | MiniSatCS | 886.007 | 21.711 | 20% | 0% |
| | | Ternary+10xCBD | 92.91% | 78% | MiniSatCS | 342.097 | 4.742 | 5% | 2% |
| | | BinMask | 98.94% | 86% | MiniSatCS | 2595.032 | 3600.001 | 70% | 2% |
| | | BinMask+CBD | 96.88% | 89% | MiniSatCS | 0.664 | 0.028 | 0 | 70% |
| | | | | | MiniSat | 225.861 | 18.761 | 0 | 70% |
| | | | | | Z3 | 146.567 | 0.997 | 0 | 70% |
| | | | | | RoundingSat | 33.922 | 0.702 | 0 | 70% |
| 0.3 | conv-small | Ternary (wd0) | 94.72% | 80% | MiniSatCS | 186.935 | 0.105 | 5% | 30% |
| | | Ternary (wd1) | 89.53% | 93% | MiniSatCS | 0.005 | 0.002 | 0 | 35% |
| | | BinMask | 94.31% | 94% | MiniSatCS | 0.001 | 0.001 | 0 | 52% |
| | | | | | MiniSat | 0.060 | 0.024 | 0 | 52% |
| | | | | | Z3 | 0.040 | 0.040 | 0 | 52% |
| | | | | | RoundingSat | 0.021 | 0.031 | 0 | 52% |
| | | | | | MiniSat-CN | 0.034 | 0.008 | 0 | 52% |
| | conv-large | Ternary | 96.89% | 91% | MiniSatCS | 2828.479 | 3600.001 | 78% | 0% |
| | | Ternary+CBD | 81.33% | 80% | MiniSatCS | 0.034 | 0.020 | 0 | 42% |
| | | | | | MiniSat | 173.877 | 23.527 | 0 | 42% |
| | | | | | Z3 | 2.093 | 1.840 | 0 | 42% |
| | | | | | RoundingSat | 2.941 | 1.642 | 0 | 42% |
| | | BinMask | 98.88% | 82% | MiniSatCS | 2442.698 | 3600.001 | 65% | 5% |
| | | BinMask+CBD | 96.26% | 87% | MiniSatCS | 0.005 | 0.005 | 0 | 52% |
| | | | | | MiniSat | 0.242 | 0.045 | 0 | 52% |
| | | | | | Z3 | 0.530 | 0.540 | 0 | 52% |
| | | | | | RoundingSat | 0.088 | 0.104 | 0 | 52% |
| | | | | | MiniSat-CN | 0.388 | 0.076 | 0 | 52% |

weight decay of $1\mathrm{e}{-7}$, except for the undefended `conv-large` networks on CIFAR10 which have a larger weight decay of $2.5\mathrm{e}{-6}$ due to the low sparsity under the default setting.

We consider the following questions for ablation study:

**Q: Does native handling of reified cardinality constraints always facilitate the SAT solving?**

**A:** Yes. We compare the solving time of `MiniSatCS`, `MiniSat`, `Z3`, and `RoundingSat` on both network architectures trained on both datasets. The sequential counters encoding is used for `MiniSat`, and we also evaluate `MiniSat-CN` that uses the cardinality networks encoding on a few cases, but it

Table 4: Comparison of methods on 40 randomly chosen CIFAR10 test images with solving time limit of 3600 seconds.

| $\epsilon_{train}$ | Network Architecture | Training Method | Test Accuracy | Sparsity | Solver | Mean Solve Time | Median Solve Time | Timeout | Verifiable Accuracy |
|---|---|---|---|---|---|---|---|---|---|
| 0 | conv-small | Ternary | 54.78% | 82% | MiniSatCS | 0.267 | 0.006 | 0 | 0% |
| | | | | | MiniSat | 327.303 | 50.036 | 7% | 0% |
| | | | | | Z3 | 411.638 | 117.604 | 5% | 0% |
| | | | | | RoundingSat | 0.361 | 0.098 | 0 | 0% |
| | | BinMask | 55.22% | 79% | MiniSatCS | 0.003 | 0.003 | 0 | 0% |
| | | | | | MiniSat | 3.981 | 3.577 | 0 | 0% |
| | | | | | Z3 | 0.590 | 0.376 | 0 | 0% |
| | | | | | RoundingSat | 0.081 | 0.077 | 0 | 0% |
| | conv-large | Ternary | 69.25% | 89% | MiniSatCS | 823.370 | 1.860 | 20% | 0% |
| | | BinMask | 67.46% | 94% | MiniSatCS | 300.404 | 3.201 | 5% | 0% |
| | | BinMask+CBD | 63.18% | 88% | MiniSatCS | 1.415 | 0.048 | 0 | 0% |
| | | | | | MiniSat | 168.079 | 69.471 | 0 | 0% |
| | | | | | Z3 | 3515.386 | 3600.121 | 92% | 0% |
| | | | | | RoundingSat | 19.162 | 0.858 | 0 | 0% |
| 8/255 | conv-small | Ternary | 32.59% | 95% | MiniSatCS | 0.002 | 0.002 | 0 | 15% |
| | | BinMask | 37.75% | 96% | MiniSatCS | 0.001 | 0.002 | 0 | 18% |
| | | | | | MiniSat | 0.070 | 0.082 | 0 | 18% |
| | | | | | Z3 | 0.050 | 0.052 | 0 | 18% |
| | | | | | RoundingSat | 0.033 | 0.043 | 0 | 18% |
| | conv-large | Ternary | 34.60% | 94% | MiniSatCS | 241.572 | 0.047 | 5% | 10% |
| | | | | | RoundingSat | 516.767 | 1.154 | 12% | 10% |
| | | BinMask | 53.91% | 88% | MiniSatCS | 206.037 | 0.052 | 5% | 0% |
| | | | | | RoundingSat | 301.062 | 0.546 | 7% | 0% |
| | | BinMask+CBD | 38.75% | 87% | MiniSatCS | 0.009 | 0.011 | 0 | 20% |
| | | | | | MiniSat | 0.224 | 0.267 | 0 | 20% |
| | | | | | Z3 | 0.768 | 0.795 | 0 | 20% |
| | | | | | RoundingSat | 0.218 | 0.126 | 0 | 20% |

is not consistently better than `MiniSat`. Our solver `MiniSatCS`, extended from `MiniSat` with native handling of reified cardinality constraints, delivers a speedup of mean solving time by a factor of between 1.35 to 5104.94 times compared all other solvers in all cases, and the average speedup is 40.21. The speedup of median solving time of `MiniSatCS` compared to others is 8.93 to 75289.76. Note that although the encoding complexity of `MiniSat-CN` is $O(n \log^2 b)$ which is asymptotically better than $O(nb)$ of `MiniSat`, the low cardinality bounds in our networks make such asymptotic comparison inaccurate. Also note that `MiniSatCS` is constantly faster than all other solvers, and no solver is constantly the second fastest (`RoundingSat` is usually faster than Z3 and `MiniSat`, but it is slower than Z3 in the `Ternary+CBD` setting with the adversarially trained `conv-large` network on MNIST).

**Q: How fast do BinMask networks verify compared to ternary quantization networks?**

**A:** For the `conv-small` networks, BinMask networks verify significantly faster than ternary networks with similar sparsity, especially in the undefended training setting. For the `conv-large` networks without the CBD loss, neither of them constantly verifies faster than the other. We note that for the `conv-large` networks, BinMask still produces more balanced layer-wise sparsities, which are, for example, [9% 21% 21% 28% 90% 87% 41%] and [84% 61% 65% 61% 87% 86% 70%] for the undefended ternary and BinMask networks on MNIST respectively. Their verification speeds are both slow because the high cardinality bounds dominate verification complexity, which are 115.3 and 119.4 on average for each neuron in the two networks respectively.

**Q: How accurate are BinMask networks compared to ternary quantization networks?**

**A:** Interestingly, ternary quantization networks have slightly higher test accuracy in most of the undefended training cases, but BinMask networks have both higher test accuracy and verifiable accuracy when trained adversarially. We highlight the comparison of `Ternary (wd0)`, `Ternary (wd1)`, and `BinMask` on the adversarially trained `conv-small` network in Table 3. The `wd0` ternary

network has a weaker weight decay to match the accuracy with the BinMask network, but it verifies much slower and has lowered verifiable accuracy. The `wd1` ternary network is trained with a stronger weight decay to match the sparsity of the BinMask network, but it has much lower test accuracy and also verifies slower. Our results suggest that BinMask not only reduces verification complexity, but also regularizes model capacity to make it more robust.

**Q: Does the CBD loss reduce cardinality bound and speed up verification for ternary quantization networks?**

**A:** Yes, but less effectively. We train `conv-large` networks on MNIST with ternary quantization and CBD loss as the `Ternary+CBD` networks shown in Table 3. The CBD loss induces denser networks with ternary quantization, and the weight decay of `Ternary+CBD` networks is increased by five times compared to `Ternary` networks to maintain comparable sparsity. The average cardinality bound of the undefended `Ternary+CBD` network is 4.3, compared to 115.3 of the `Ternary` network. Although the `Ternary+CBD` and `BinMask+CBD` networks achieve similar average cardinality bounds (4.3 vs 3.8), the ternary network has much higher maximal cardinality bound (146.3 vs 22.3). Therefore, the verification time of `Ternary+CBD` is significantly improved over `Ternary` but is still longer than that of the `BinMask+CBD` network. `Ternary+10xCBD` is obtained by increasing the CBD loss coefficient $\eta$ by ten times on `Ternary+CBD`, which has a lower average cardinality bound of 3.2, but the maximal cardinality bound is not decreased (157.7) and its test accuracy is much worse. The adversarially trained `Ternary+CBD` network has a lower maximal cardinality bound of 73.1, which also verifies faster. Ternary quantization networks with the CBD loss suffer from a larger decline of test accuracy compared to BinMask networks. Our results suggest that lower cardinality bound reduces verification complexity, and the BinMask formulation makes it easier to optimize for lower cardinality bounds.

**Q: Do other solvers benefit from more balanced layer-wise sparsities and/or lower cardinality bounds?**

**A:** Yes. We evaluate `MiniSat`, `Z3`, and `RoundingSat` on a relatively easy to verify ternary network (undefended `conv-small` on CIFAR10 in Table 4). The results show that all the solvers achieve significant speedup on the corresponding BinMask network, although `Z3` benefits more from BinMask than `MiniSat` and `RoundingSat`. We also try `MiniSat` and `Z3` on the easiest-to-verify `conv-large` network trained with only BinMask (i.e., the one adversarially trained on CIFAR10), but `MiniSat` fails due to out of memory error, and `Z3` always exceeds the one hour time limit (data not shown in the table). With `BinMask+CBD` they both verify much faster. `RoundingSat` also benefits from `BinMask` and `BinMask+CBD` in this setting as shown in the table. Note that the ternary network has lower test accuracy and higher overall sparsity, but still verifies slower for all the solvers. Our results suggest that our two strategies, which are inducing more balanced layer-wise sparsity and lower cardinality bounds, both reduce the complexity of the verification problem and facilitate all the solvers that we have considered.