[Reviews · NeurIPS 2020]

Review 1

Summary and Contributions: As promised by the title, the paper presents an algorithm for the exact verification of BNNs. The BNN and the robustness specification are encoded as a set of Boolean clauses. A SAT solver is used; the standard SAT solver is modified to handle reified cardinality constraints. Empirical results show that this leads to considerable decrease in computation time (2 to 3 orders of magnitude). A second contribution is the training method and the enforcement of sparsity.

Strengths: The problem setting itself is not new. The key contribution is the modification to the SAT solver which results in significant decrease in training time. The paper hypothesizes that imbalance in the layer-wise sparsity slows down the SAT solver; some empirical evidence is provided, and the paper then proposes a novel design method to enforce it. A variation of gradient canceling is proposed to induce robustness.

Weaknesses: All the empirical results are for small examples (CIFAR-10) raising the question of scalability. Added post-discussion: The authors' acknowledge this issue in the feedback, but provide no potential approaches to lowering complexity / enabling scalability.

Correctness: No significant issues.

Clarity: Yes

Relation to Prior Work: Seems to be well-positioned wrt SoA

Reproducibility: Yes

Additional Feedback:


Review 2

Summary and Contributions: The manuscript studies the problem of verifying BNNs using SAT solvers. The manuscript introduces a methodology that is based on i) modifying an existing SAT solver (i.e., MiniSAT) to natively handle reification of cardinality constraints, and train solver friendly BNNs for verification by the means of lowering their bias. Experimentally, the introduced method (i.e., EEV) outperforms SAT solvers with sequential counters by Sinz et al. 2005, and a SMT solver (i.e., z3) over MNIST and CIFAR10 datasets.

Strengths: The main strengths of the work are i) the potential efficiency of adapting existing SAT solvers to handle BNN verification problems more efficiently compared to naive cardinality encodings, and ii) the training methodology for balanced sparsity in BNNs.

Weaknesses: The main weaknesses of the work include i) missing experimental baseline comparisons and ii) missing discussion of related works in BNN and SAT literatures on similar problems. For these reasons which will be explained in detail below, the manuscript is not ready for a publication at NeurIPS in its current form.

Correctness: Comment 1: The main contribution of the paper is the extension of an existing SAT solver (i.e., MiniSAT) to a SAT solver that can handle (reified) cardinality constraints. There exists successful solvers already in the pseudo-Boolean (some of which DIRECTLY extend **MiniSAT**) and constraint programming research that can handle cardinality constraints without the use of auxiliary variables/constraints. The manuscripts unfortunately completely skips areas of research that are highly relevant. Unfortunately the manuscript neither discusses why they have not been used for comparison, or use them as meaningful baselines.

Clarity: The paper is well-written aside from important missing related work and meaningful baseline comparisons.

Relation to Prior Work: Comment 2: In the SAT research, there are more compact and efficient encodings for encoding cardinality constraints compared to sequential counters by Sinz et al. 2005 (e.g., cardinality networks by Asin et al. 2011). In the planning with BNNs research by Say et al 2020, cardinality networks have been extended to reified cardinality constraints, and have been shown to perform faster compared to sequential counter-based encodings. Comment 3: There also exists a (preliminary) work on training BNNs with constraint programming that uses low amounts of data by Icarte et al. 2019 - which you can use as a reference for training sparse BNNs.

Reproducibility: No

Additional Feedback: Additional References: Asín, Roberto and Nieuwenhuis, Robert and Oliveras, Albert and Rodríguez-Carbonell, Enric. Cardinality Networks and Their Applications. Theory and Applications of Satisfiability Testing - SAT 2009, pp. 167-180, 2009. Icarte, Rodrigo Toro and Illanes, Leon and Castro, Margarita P. and Cire, Andre A. and McIlraith, Sheila and Beck, Christopher J. Training Binarized Neural Networks using MIP and CP. Principles and Practice of Constraint Programming. pp. 40-417, 2019. Say, Buser and Sanner, Scott. Compact and Efficient Encodings for Planning in Factored State and Action Spaces with Learned Binarized Neural Network Transition Models. Artificial Intelligence, 285, 103291, 2020. **ATFER REBUTTAL** Thank you for your responses and the additional set of experiments. Below, please find my recommendations for your paper. - Please consider including the results presented during rebuttal into the final version of the paper (i.e., this makes the paper stronger). - Please remove the claim that the paper is the first to handle reified constraints of the form [\sum_i x_i >= b] <-> y natively. - Please note there already exists modelling languages and solvers (i.e., ILP, CP, pseudo Boolean optimization etc.) that achieves a native encoding for reified constraints in the size of the BNN. - Please consider releasing your code.


Review 3

Summary and Contributions: The paper addresses the problem of verifying binary neural networks (BNN) and of learning verifiable BNNs. To this end, it makes the following contributions: - A extension of MiniSat that handles reified cardinality constraints that arise in BNN verification. - Balanced sparsification and cardinality bound decay for training verifiable BNNs. - Improved PGD attacks for adversarial training of BNNs.

Strengths: + The speed of verifying BNNs over real-valued neural networks is significantly better. + Each of the contributions made in the paper seems to help further speed-up verification. + The paper provides a good high-level overview of the approach.

Weaknesses: - Some aspects of the proposed approach require more explanation (please see the "Clarity" section for details). - The accuracy of BNNs is worse than the real-valued networks (please see the "Correctness" section for details).

Correctness: - In section 4.1, how is the proposed encoding different from [42]? How does the method of [42] compare with the proposed approach in terms of speed? - In section 3.2, how much does the accuracy suffer due to the changes made in the first and the last layer in order to help with the SAT formulation of verification? - In Table 2, PGD accuracy has no noticeable pattern wrt $\eta$. Any thoughts on why that might be the case? - The accuracy of BNNs is lower (sometimes significantly lower) than the real-valued networks. Given the faster inference and verification time of BNNs, it would be interesting to see if larger BNNs can be used that still offer a speed-up but are much closer in terms of accuracy. - In the "reject option" paragraph of the experiments section, I'm not sure I understand why the encoding would not be straightforward in real-valued networks. See, for example, the supplementary of [9], which talks about CNF type formulas for combining linear specifications.

Clarity: - While the paper provides a good high-level overview of the proposed approach, it would be difficult to reproduce the results. The paper does state that the code will be open sourced, which would be essential for future research based on this submission. - Why does sparsity help speed-up verification in the case of BNNs? Is it because the number of variables that the SAT solver needs to handle reduces? Or that propagation is easier? - In section 5.1, why does the reformulation in equation (5) address the two drawbacks of ternary quantization mentioned at the start of the section?

Relation to Prior Work: To my knowledge, relevant prior work has been cited. The relationship between the proposed encoding and [42] is not clear. A direct comparison of the proposed method and [42] is also missing.

Reproducibility: No

Additional Feedback: Post-rebuttal comment: I have read the rebuttal and the other reviews. While I agree with reviewer 2 that there exist other alternatives to the proposed extension, I still believe there is merit in this work as it applies the extended SAT formulation to binary neural network verification. I believe the claims of the paper can be restated in light of reviewer 2's comments in the final version of the paper, and hence, still support acceptance.


Review 4

Summary and Contributions: The paper aims to scale up formal verification of binarized neural networks. The authors push on two fronts: 1) tweaking a SAT solver to be more specialized in solving CNF encodings of BNNs, and 2) tweaking the learning of BNN to reduce cardinality bounds (which authors claim are the biggest bottleneck in current approaches) using new masking and regularization strategies.

Strengths: Well written, it was easy/clear to read. The speedup in the experiments are quite impressive. The bookkeeping for CDCL seems very natural/intuitive, so it is nice to see a simple solution that works. Since most of the clauses are derived from cardinality constraints, the authors propose to do some bookkeeping to detect when a cardinality constraint only has a unique solution, and then directly fill in that unique solution. The authors also propose to limit the size of the cardinality constraints by regularizing/penalizing high cardinalities directly during training. Also appreciated that the authors conducted an ablation study in the appendix to study the individual effects of each method. Overall the problem of verifying BNNs is quite relevant/important, and the authors propose a variety of improvements that all seem quite intuitive, well motivated and are backed up by the experiments.

Weaknesses: Section 6 seems a bit out of place in a paper titled "Efficient Exact Verification of Binarized Neural Networks", considering its focus on training more robust BNNs. The section deviates from the overall theme of improving verification efficiency.

Correctness: Yes, to the best of my judgment.

Clarity: Yes, clear and well written.

Relation to Prior Work: Yes, the authors present three techniques (bookkeeping cardinality constraints in CDCL, binmask, and regularizing cardinality bounds that are all novel to the best of my judgment.

Reproducibility: Yes

Additional Feedback: The parameter tau=5 in the regularization seems like it would be important to the verification efficiency and model accuracy tradeoff. Did you try other values for tau? For the experiments in Section 4, do the other baselines use a variable ordering that considers variables derived from the same cardinality constraint in order? If not, could this simple tweak be added to also get close to the same benefits as the bookkeeping technique proposed in MiniSatCS? Minor complaint -- since the code will only be provided after reviewing, it is perhaps a little unfair to check off training/evaluation code under "ML Reproducibility -- Code". I have read the rebuttal, my opinion has not changed.

[Author Response · NeurIPS 2020]

We thank the reviewers for providing those helpful comments, especially during the challenging time this year. Below are our responses to individual reviewers:

**Reviewer #1**

> *All the empirical results are for small examples (CIFAR-10) raising the question of scalability.*

We understand the concern. We note that previous complete verification methods (for either binarized or real-valued neural networks) report results only for similarly sized networks and datasets and the results show that our work significantly improves the scalability of complete BNN verification in comparison with all previous methods.

**Reviewer #2**

> *There exists successful solvers already in the pseudo-Boolean and constraint programming research that can handle*
> *cardinality constraints without the use of auxiliary variables/constraints.*

BNN verification requires solving reified cardinality constraints of the form $y \leftrightarrow (\sum_{i=1}^{n} x_i \geq p)$, NOT cardinality constraints of the form $\sum_{i=1}^{n} x_i \geq p$. We are the first to support reified cardinality constraints natively in the SAT solver. We cite previous work on SAT solver support for cardinality constraints (Liffiton et al. [35]) and are happy to cite additional research as suggested by the reviewer. However, none of this research can solve BNN verification problems due to their requirement for reified constraints and is therefore not a comparison baseline for our technique. We do compare with Z3, which supports general pseudo-Boolean constraints including reified cardinality constraints, specifically via on-demand compilation into sorting circuits. The results show that our native support for reified cardinality constraints significantly outperforms Z3.

> *In the planning with BNNs research by [1], cardinality networks have been extended to reified cardinality constraints,*
> *and have been shown to perform faster compared to sequential counter-based encodings.*

Instead of natively supporting reified cardinality constraints, [1] encodes such constraints using cardinality networks. This encoding requires $O(n \log_2^2 p)$ auxiliary variables/clauses. Our research, in contrast, modifies the SAT solver to natively handle reified cardinality constraints with no auxiliary variables/clauses. We implemented the encoding of [1] and tested it on three MNIST networks: `MNIST-MLP`, `conv-small`, and `conv-large` ($\eta = 5e - 4$). The results show that, with native support for reified cardinality constraints, our solver delivers speedups of 150.5x, 55.3x, and 90.2x compared to the encoding of [1] in three cases respectively. We are happy to include related discussion and results in the next version. Note that [1] deals with planning, not BNN verification, and was published after the NeurIPS deadline.

> *There are more compact and efficient encodings for encoding cardinality constraints compared to sequential counters.*

We use sequential counters only for comparison with other BNN verification research (e.g. Narodytska et al. [43]), which uses sequential counters. We advocate native support for reified cardinality constraints in the SAT solver, not encoding reified constraints using sequential counters, sorting networks, or cardinality networks as in [1], and the results show that our approach significantly outperforms all of these encodings.

> *The main contribution of the paper is the extension of an existing SAT solver (i.e., MiniSAT) to a SAT solver that can*
> *handle (reified) cardinality constraints.*

We make several main contributions, including efficient SAT solving, training solver-friendly BNNs via inducing two properties, and training robust BNNs via adaptive gradient cancelling. Their working together is required to obtain the performance numbers in the paper — while our novel direct support for reified cardinality constraints is a critical contribution with potential applications to other SAT problems, it is one of three main contributions in the paper.

> *There also exists a (preliminary) work on training BNNs with constraint programming that uses low amounts of data*
> *by Icarte et al. 2019 - which you can use as a reference for training sparse BNNs.*

Icarte et al. 2019 focuses on generalizability in few-shot learning, which is a different setting than ours. Specifically, they train BNNs of 0, 1, or 2 hidden layers with 16 neurons each, on no more than 10 training examples per class of MNIST with a two-hour time limit, while our research works with larger networks and more data. We are happy to reference and discuss Icarte et al. 2019 in the next version.

**Reviewer #3**

> *In section 4.1, how is the proposed encoding different from [42] and how do they compare in terms of speed?*

We use quantized floating point input (vs binary input in [42]), constrain activations to $\{0, 1\}$ (vs $\{-1, 1\}$ in [42]), and include a modified BN in the last layer (vs no last layer BN in [42]). We use [43], which is an improved version of [42], as a baseline for comparison and have demonstrated significant speedup (Figure 1). We will clarify in the next version.

**Reviewer #4**

> *Did you try other values for $\tau$?*

We tried multiple values for $\tau$ (such as 3, 5, 10). $\tau = 5$ was a good choice that enables fine control of the accuracy-speed tradeoff by tuning $\eta$. Thanks for the feedback, and we will clarify and discuss in the next version.

**References**

[1] Buser Say and Scott Sanner. Compact and efficient encodings for planning in factored state and action spaces with learned binarized neural network transition models. *Artificial Intelligence*, page 103291, 2020.


[Meta-Review · NeurIPS 2020]

The paper was assessed as a high quality work by most of the reviewers, contributing fast methods for robustness verification of binary neural networks and training robust binary networks. The points of strong criticism were positioning of the contribution wrt to the constraint programming methods. Since one of the main claimed contributions is the speed-up, it was questioned whether such a speed-up can be obtained by just existing methods / solvers. In particular, L168: "We present the first extension to handle the reified cardinality constraints" was criticized. The arguments of the discussion clarified that in modern pseudo-Boolean solvers the same (resp. extended) CDCL methods are used to handle constraints. Cardinality constraints and more generally linear inequality constraints can be handled natively. In particular [Elffersand and Nordstrom "Divide and Conquer: Towards Faster Pseudo-Boolean Solving" 2018] describes propagating such constraints directly without any auxiliary variables, which in the implementation may actually efficiently specialize for 0/1 coefficients. This is more than authors say in L135 about PB solvers. The argument then was that reified cardinality constraints are not significantly more complex as they can be encoded as two inequality constraints e.g. as follows: y = [ \sum_i x_i +b >= 0] can be encoded as y < \sum_i x_i +b n*y >= \sum_i x_i +b -1, where n is the number of variables R2 would expect this to be the baseline and would expect the speed-up measured against this baseline would be not as outstanding. However other reviewers agreed, that from the point of view of the current state of the art on BNN verification, the comparison with previously applied methods is valid. A claim of speeding-up over applicable solvers in general would require more thorough discussion of methods and benchmarking. The authors should address all issues raised by reviewers in the final version to the best of their judgement. In particular include comparisons presented in the rebuttal, relate more densely to the literature on constraint programming and refine the claimed contributions, focusing on the BNN application. As a possible extension, optimization rather than SAT formulation may offer solving for the largest eps such that the verification still holds.